# Malignant Clinical Course of “Proliferative” Ovarian Struma: Diagnostic Challenges and Treatment Pitfalls

**DOI:** 10.3390/diagnostics12061411

**Published:** 2022-06-07

**Authors:** Aleksandra Asaturova, Alina Magnaeva, Anna Tregubova, Vlada Kometova, Yevgeniy Karamurzin, Sergey Martynov, Yuliya Lipatenkova, Leila Adamyan, Andrea Palicelli

**Affiliations:** 1FSBI “National Medical Research Centre for Obstetrics, Gynecology and Perinatology Named after Academician V.I.Kulakov” of the Ministry of Health of the Russian Federation, Bldg 4, Oparina Street, Moscow 117513, Russia; a_magnaeva@oparina4.ru (A.M.); a_tregubova@oparina4.ru (A.T.); v_kometova@oparina4.ru (V.K.); s_martynov@oparina4.ru (S.M.); sol-juli@yandex.ru (Y.L.); l_adamyan@oparina4.ru (L.A.); 2Mediclinic Middle East, Dubai Healthcare City, Bldg 37, Dubai P.O. Box 505004, United Arab Emirates; ykaramur@gmail.com; 3Pathology Unit, Azienda Unità Sanitaria Locale—IRCCS di Reggio Emilia, 42123 Reggio Emilia, Italy; andrea.palicelli@ausl.re.it

**Keywords:** struma ovarii, peritoneal strumosis, well differentiated follicular carcinoma arising in struma ovarii

## Abstract

Struma ovarii (SO) is a monodermal teratoma predominantly composed of thyroid tissue (TT) showing benign, “proliferative”, or malignant histology. By imaging, a 38-year-old patient with lower backache revealed a 6.2-cm vertebral lesion (L5). Core biopsy showed well-differentiated TT without features of papillary carcinoma. A 3.5-cm left ovarian mature teratoma (lacking TT) and peritoneal nodules (showing well-differentiated TT) were also identified and surgically removed. Thyroid ultrasound and cytological examination resulted negative. Four years before, left ovarian cystectomy was performed for a histologically “proliferative” SO. According to the malignant clinical course and WHO classification, this case was overall reassessed as a recurring well-differentiated follicular carcinoma arising in SO (WD-FC-SO), despite lacking malignant histological features in any specimens. Immunophenotype: TTF-1+/PAX-8+/thyroglobulin+/CK7+/chromogranin-/synaptophysin-/inhibin-/calretinin-/HNF1B-; Ki-67 index < 5%. Polymerase chain reaction analysis resulted negative for *BRAF^V^*^600*E*^ mutation. The patient refused further treatments, without recurrence after 17 months. The clinical behavior of SO may be unpredictable. Histologically benign or proliferative strumas extraordinarily metastasize, while SO with malignant features may not recur. The exceptional evidence of peritoneal implants of well-differentiated TT (peritoneal strumosis) in patients with histologically benign SO represents a metastasis of WD-FC-SO (like in our case). A multidisciplinary approach including clinical, laboratory, radiologic, and histopathological data is required.

## 1. Introduction

Struma ovarii is a monodermal ovarian teratoma predominantly composed of thyroid tissue; it is a rare neoplasm, comprising 1% of all ovarian tumors and 3% of ovarian teratomas [1,2,3]. The tumor usually presents as an incidental pelvic mass with or without abdominal pain. Uncommon clinical manifestations may include hyperthyroidism (<10%), dropsy, and Meigs’ syndrome [1]. Emerging evidence indicates that the histological features and growth patterns of struma ovarii do not seem to correlate with the biological behavior. For instance, a morphologically benign struma ovarii may be associated with extraovarian spreading, known as peritoneal strumosis [4,5]. This condition in which both the peritoneal implants and the primary ovarian tumor are histologically bland is now regarded as metastasis from a well differentiated follicular carcinoma arising in struma ovarii. The phenomenon of extraovarian spreading is peculiar for several reasons. First, the mechanisms of dissemination are not well understood, although accumulating data suggest that molecular alterations, such as *ALK*, *EGFR* and *BRAF* mutations, may contribute to tumor spread [6,7]. Second, the treatment approaches are not standardized and may currently include various doses of radioactive ablation with I-131 and thyroidectomy [4,8,9]. Moreover, the differential diagnosis includes entities such as neuroendocrine tumors and metastases by adenocarcinomas of various origin (such as follicular thyroid carcinoma), making the diagnosis a real challenge for pathologists and clinicians [10,11].

We report a clinical case of a morphologically “proliferative” ovarian struma that subsequently progressed with the development of peritoneal and osseous metastases four years after tumor resection.

## 2. Case Description

A 38-year-old woman with a history of struma ovarii was admitted to the gynecological surgery department due to an ovarian tumor recurrence. Indeed, she had previously undergone a left ovarian cystectomy performed during a cesarean section in 2016.

In May 2020, the patient was examined for lower back pain. Magnetic resonance imaging revealed a lesion involving the fifth lumbar (L5) vertebral body and measuring 6.2 × 3.4 × 2.7 cm; it was associated with vertical fracture line, spinal canal narrowing, left neural foramen reduction, and nerve root compression. Then, a Technetium 99m Sestamibi whole-body scan demonstrated bone tissue remodeling in the lumbar column.

A core biopsy of the vertebral lesion showed clusters of well-differentiated thyroid follicles expressing thyroglobulin; the typical architectural and cytological features of thyroid papillary carcinoma (clear nuclei with nuclear grooves and pseudoinclusions) were lacking (Figure 1).

Percutaneous image-guided vertebroplasty was performed due to the risk of an L5 pathologic fracture and for severe back pain.

Positron emission tomography—computed tomography (PET-CT) scans showed not only an osteolytic lesion in the L5 vertebral body but also a left ovarian mass with non-homogeneous density and fat, highly suggestive of teratoma (Figure 2).

Taking into account the results of the lumbar core biopsy, ultrasound examination (US) of the thyroid gland was performed to search for a primary thyroid carcinoma. It revealed a small nodule, which resulted with a benign lesion on fine-needle aspiration cytology examination (Bethesda diagnostic category II). No hyperthyroidism symptoms or abnormal levels of thyroid hormones or thyroid-stimulating hormone (TSH) were identified.

Given the abovementioned results and the patient’s past medical history, a metastasis from ovarian neoplasm was suspected, and the woman was admitted to the surgery department of the *National Medical Center for Obstetrics, Gynecology and Perinatology named after V.I. Kulakov* in September 2020.

The preoperative pelvic US scans demonstrated a left ovarian, solid mass of 3.5 × 3.4 × 3.0 cm; the tumor was hyperechoic and showed clear smooth contours without vascularization zones (Figure 3).

We compared these results with the findings of the US scan performed at the time of the left ovarian cystectomy for ovarian struma in 2016. The latter exam had demonstrated a solid and cystic ovarian tumor of 7.5 × 5.0 × 3.5 cm with heterogeneous solid components, increased echogenicity, and vascularization zones at the periphery (Figure 4).

No peritoneal deposits had been found during surgery in 2016.

In 2020, our laparoscopic procedure revealed an ovarian mass and solid implants involving the serosa of the greater omentum and sigmoid colon (Figure 5); laparoscopic left oophorectomy, peritoneal biopsies, omentectomy, and excision of the sigmoid serosal lesions were performed.

On gross examination, the surgical specimens comprised an ovarian solid and cystic tumor measuring 3.5 × 2.0 × 1.0 cm; the cystic parts were filled with hair shafts, while fat tissue and cartilage were identified by cutting the solid component. Moreover, four macroscopic peritoneal implants were found on the omentum and sigmoid serosa, ranging in size from 0.3 to 1.0 cm.

On histological examination, the left ovarian tumor included mature elements (keratinized multilayered squamous epithelium, cutaneous adnexal structures, respiratory epithelium, fibroadipose tissue, mature cartilage, and glia). No evidence of thyroid tissue was found, despite the total embedding of the submitted specimen. At the same time, the peritoneal implants demonstrated thyroid follicles of various sizes lined by cells lacking the typical histological features of papillary carcinoma.

The morphology of these metastases was compared to that of the original struma ovarii, which had been surgically removed in 2016. The previous slides were retrieved from our histological archive and carefully reviewed. The primary ovarian neoplasm demonstrated macro- and micro-follicles containing colloid, including areas of densely packed follicles; the typical architectural and cytological features of papillary carcinoma were lacking, as well as other clearly malignant features typical of follicular carcinoma, including lymphovascular or capsule invasion (Figure 6).

Due to these histologic features, a diagnosis of cellular (so-called “proliferative”) struma of the left ovary was rendered.

On immunohistochemical examination, we demonstrated that the expression of the following immunohistochemical markers was identical in either the primary ovarian tumor or the metastases. Indeed, the tumor cells were diffusely positive for TTF-1 (clone 8G7G3/1, mouse monoclonal, RTU; Cell Marque, Rocklin, CA, USA), PAX-8 (clone MRQ-50, mouse monoclonal, RTU; Cell Marque), thyroglobulin (clone EPR 9730, rabbit monoclonal, dilution 1:500; Abcam, Cambridge, UK) and CK7 (clone SP52, rabbit monoclonal, RTU; Ventana Medical Systems, Oro Valley, AZ, USA), but negative for chromogranin (clone LK2H10, mouse monoclonal, RTU; Ventana), synaptophysin (clone SP11, rabbit monoclonal, RTU; Ventana), inhibin-A (clone R1, mouse monoclonal, RTU; Cell Marque), calretinin (clone SP65, rabbit monoclonal, RTU; Ventana) and HNF1B (clone beta, mouse monoclonal, dilution 1:100; Abcam). The Ki-67 (clone 30-9, rabbit monoclonal, RTU; Ventana) labeling index was below 5% (Figure 6); the tumor cells also tested negative for *BRAF^V^*^600*E*^ mutation on real-time polymerase chain reaction analysis.

Due to the clinical, histological, immunohistochemical, and molecular findings, the final diagnosis of a well-differentiated follicular carcinoma arising in struma ovarii was made. The patient had a child but still wanted to preserve her fertility, and she also refused the proposals of total thyroidectomy and radioactive iodine therapy; symptomatic treatment with the monoclonal antibody denosumab was carried out in order to decrease bone resorption. The 17-months follow-up showed no recurrence.

## 3. Discussion

Approximately 15% of mature cystic teratomas contain thyroid tissue, although the term “struma ovarii” must be used only when thyroid tissue is predominant or sole. Struma ovarii accounts for 2–5% of all mature cystic teratomas; malignant transformation occurs in 5–10% of the cases, and metastases appear in 5–23% of malignant struma ovarii [12,13]. This entity is of considerable interest because of its unique features. Foremost, the histological appearance does not accurately correlate with biological behavior. Somewhat paradoxically, some morphological criteria characterizing the malignant nature of thyroid gland neoplasms cannot be always applied to struma ovarii in terms of the prediction of extraovarian dissemination. While cases showing the typical cytological and architectural features of papillary or poorly-differentiated carcinomas of the thyroid were described to arise from struma ovarii, the diagnostic criterium of capsular invasion for follicular carcinomas of the thyroid may not be applicable in struma ovarii lacking a tumor capsule. In addition, other signs of malignancy (like lymphovascular invasion) could not be detectable by histologic examination, as in our case. Indeed, struma ovarii without any evidence of architectural or cytologic atypia can metastasize to the abdominal cavity or other distant sites [12,13].

Based on the morphological features, the differential diagnosis included strumal carcinoid, ovarian clear cell carcinoma, sex cord stromal tumors, and metastasis of a thyroid gland tumor. In our case, strumal carcinoid was ruled out due to the lack of chromogranin and synaptophysin immunohistochemical expression, while the hypothesis of a sex cord tumor was rejected as to the negativity for inhibin-A and calretinin. In addition, we excluded an ovarian clear cell carcinoma as to the negative expression of HNF1B and to the positivity for TTF-1 and thyroglobulin, which confirmed the origin of the tumor from thyroid-type tissue. However, a metastasis from a thyroid gland tumor was ruled out as to the results of ultrasound examination and fine-needle aspiration cytology.

Some authors defined a subgroup of struma ovarii as histologically “proliferative” due to the presence of a compact, densely cellular proliferation of small follicles and/or hyperplastic type papillary formations; clearly malignant features should lack definition [14]. We present a case of well-differentiated follicular carcinoma arising in struma ovarii and developing peritoneal and extraperitoneal metastases four years after the initial surgery; the previously resected tumor was regarded as a “proliferative” struma because of its unusual morphology and the absence of clearly malignant histological features or peritoneal implants. Further studies are required but, currently, a “proliferative” morphology cannot be considered a clear predictor of aggressive behavior, as to the limited data [14].

Various ambiguous terminologies have been applied to define cases in which both the primary ovarian tumor and the extraovarian deposits show benign-looking thyroid tissue histology. The bland tumor morphology may raise some doubts about the opportunity to use the term “metastatic malignant struma”, which may imply the presence of clearly malignant histological features. Moreover, in our case, the subsequent peritoneal and bone metastases were even more well-differentiated, if compared to the primary left ovarian tumor. In addition, for a long time, some authors have used the terms “strumosis” [4,15,16] or “ovarian struma with extraovarian spread/dissemination” [2], which do not reflect the malignant biological behavior of the tumor and may favor consideration of the extraovarian deposits as the expression of a benign or multifocal neoplasm rather than metastases from the ovarian primary. Indeed, the current WHO classification of female genital tumors indicates that the exceptional peritoneal implants of well-differentiated thyroid tissue (peritoneal strumosis) are real metastases, allowing the re-classification of the otherwise histologically benign ovarian primary as a well-differentiated follicular carcinoma (like our case).

In the absence of extraovarian disease, the clinical behavior of benign or proliferative ovarian strumas is unpredictable, although these tumors extraordinarily metastasize or recur; thus, the same International Classification of Oncological Diseases (ICD-O) code is used for benign and malignant cases [17]. At the same time some authors suggested that it would be useful to apply the ICD-O code 1 to highlight how complicated the biological potential prediction could be for ovarian strumas [18].

It should be noted that the algorithms for the optimal treatment of malignant ovarian strumas are not sufficiently standardized due to the rarity of these tumors, the difficulties in predicting their clinical course, and the need to take into account the reproductive status of the patients. Some authors proposed the excision of the primary ovarian tumor, followed by thyroidectomy and radioactive ablation with Iodine-131, in order to eliminate unresectable metastases and favor the monitoring of thyroglobulin levels after surgery [8,9,19]. Nevertheless, it is quite difficult to accurately assess the advantages of this approach, because of insufficient data about long-term follow-up. The ovarian struma may sometimes cause hyperthyroidism or be associated with circulating TSH receptor antibodies (TSHR-Abs), representing a diagnostic challenge for clinicians [20,21,22].

In our patient, a small nodule was found in the thyroid gland but fine-needle aspiration cytology resulted in being negative for malignancy. Taking into account the results of the cytological analysis together with the patient’s refusal of total thyroidectomy, it was decided to postpone radioiodine therapy, favoring a symptomatic treatment with the fully human monoclonal antibody denosumab in order to decrease the bone resorption. Although no relapse was found after 17 months, the efficacy of this treatment will be assessed over time; however, lifelong hormone replacement therapy may cause side effects to the patients. Indeed, alternative management protocols should be further discussed to find the best approach to these rare tumors.

The clinic-pathologic features and outcomes of some previously reported cases of struma ovarii are summarized in Table 1 [2,5,13,14,23,24,25,26,27,28,29,30,31,32,33].

These findings demonstrated that benign strumas can have a malignant clinical course; Akahira J. et al. reported a histologically benign struma recurring with multiple metastases after 10 years [5]. Conversely, other cases of malignant strumas may not recur or can show long progression-free survival [13,26].

The average time to recurrence for struma ovarii can vary from some months to several years based on our literature review. In addition, the histopathological features and clinical course did not correlate with patients’ age; the median age varied from 41 to 46 years. Various metastatic sites were reported, and the localization of the metastases did not correlate with the morphological features. In our literature review, we analyzed publications including 232 benign, 41 atypical, and 134 malignant ovarian strumas [2,5,13,14,22,23,24,25,26,27,28,29,30,31,32,33]. Salpingo-oophorectomy, thyroidectomy and hysterectomy were the most common treatments. The progression-free survival did not depend on the treatment approach. The majority of patients were alive (with or without disease) at the end of follow-up. Few patients included in our review died of the disease; in nine cases, a clearly evident thyroid-type carcinoma (four anaplastic, four papillary, one follicular) was identified in the struma, while four cases were previously diagnosed as follicular adenomas arising in struma ovarii and one additional ovarian struma was histologically normal [2,23].

In our case, the ovarian struma has been retrospectively considered to harbor a histologically inapparent, well differentiated follicular carcinoma, according to the subsequent clinical course and the current WHO classification. Interestingly, our malignant struma developed during pregnancy, which may favor the growth of carcinomas arising from struma ovarii as well as of benign ovarian strumas. Estrogen is found at high levels during pregnancy, binding to receptors present in cancer cells. Moreover, recent studies revealed that human chorionic gonadotropin (hCG) can stimulate the TSH receptors of struma ovarii, potentially promoting tumor growth. Mature human granulosa cells can express functional TSH receptors, which may be involved in regulating the ovarian function. Indeed, struma ovarii in pregnancy may be challenging as regards the clinical diagnosis and treatment [34,35,36].

## 4. Conclusions

We have here reported a rare case of ovarian struma with “proliferative” histological features relapsing with bone and peritoneal metastases (peritoneal strumosis) after 4 years. As to the criteria reported in the literature, “proliferative” histology lacks malignant features by definition and, usually, it is not a predictor of poor outcome. Nevertheless, exceptional recurrences may occur, even showing more bland-looking morphological features (as in our case). Indeed, our report confirms that the clinical behavior of ovarian struma may be occasionally unpredictable. While thyroid-type carcinomas (papillary, follicular, anaplastic, etc.) may rarely arise in struma ovarii, the absence of clearly malignant histological features does not exclude a synchronous or metachronous extraovarian spread even after years. According to the current WHO classification and to our patient’s clinical course, peritoneal strumosis should be considered an expression of real metastatic deposits, allowing an “a posteriori” reclassification of the previous struma ovarii as a well-differentiated follicular carcinoma arising in struma ovarii, despite the absence of lymphovascular or capsular invasion at that time.

As tumor relapse may occur even after years (as in our case), some authors suggested a long-term follow-up for these patients. However, standardized treatment protocols have not been established, as to the rarity of recurrent strumas and their potentially indolent clinical course (at least in a group of tumors). Radioiodine therapy, oophorectomy, and thyroidectomy remain the most reasonable approaches, but our patients refused these treatments. A multidisciplinary approach is advised.

## Figures and Tables

**Figure 1 diagnostics-12-01411-f001:**
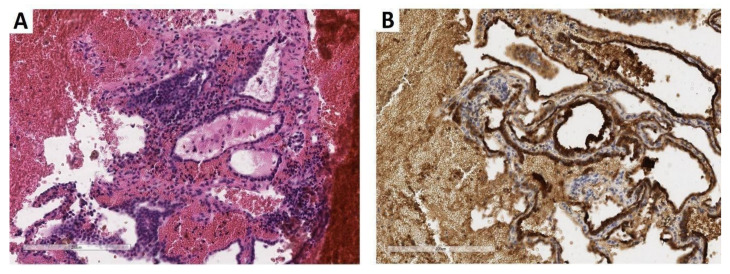
Core biopsy of the lumbar vertebral body. (**A**) The biopsy revealed thyroid-type tissue including well-differentiated follicles with colloid inside (×200; hematoxylin and eosin). (**B**) Follicular epithelial lining demonstrated marked thyroglobulin immunohistochemical expression (×200; clone EPR 9730, rabbit monoclonal, dilution 1:500, Abcam, Cambridge, UK) (previously unpublished, original photos).

**Figure 2 diagnostics-12-01411-f002:**
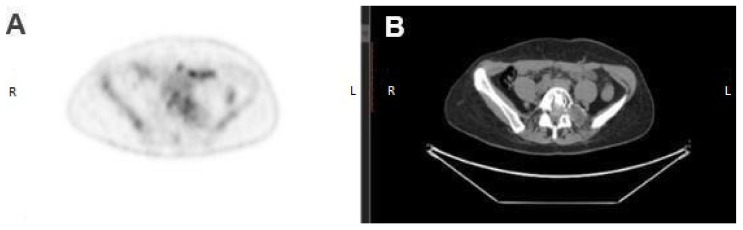
(**A**) PET image in the transverse plane, confirming the abnormal fluorodeoxyglucose uptake within the metastatic lesion localized in L5 vertebral body. (**B**) CT image at the same level showed a hypodense focus (previously unpublished, original photos).

**Figure 3 diagnostics-12-01411-f003:**
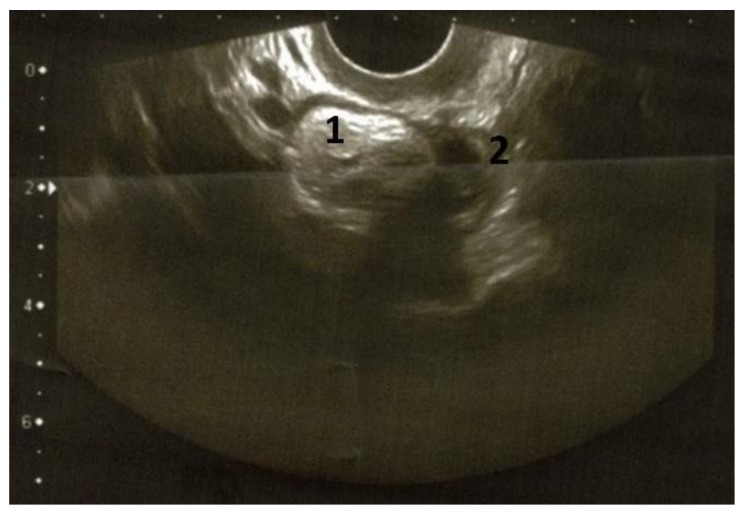
Transvaginal ultrasound examination (2020) of the left ovarian teratoma. The left ovary showed a solid tumor (3.5 × 3.0 × 3.4 cm) with increased echogenicity and clear smooth contour, without vascularization zones (1: tumor; 2: remaining ovarian tissue; previously unpublished, original photos).

**Figure 4 diagnostics-12-01411-f004:**
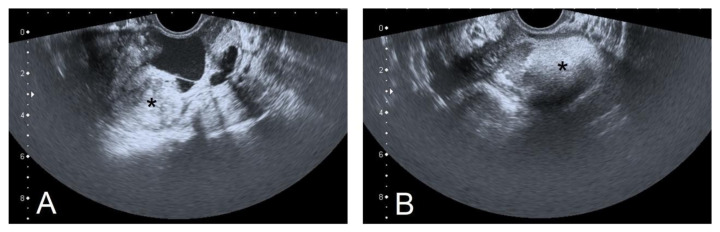
Transvaginal ultrasound examination of the left ovarian struma (2016) ((**A**,**B**); *: tumor; previously unpublished, original photos).

**Figure 5 diagnostics-12-01411-f005:**
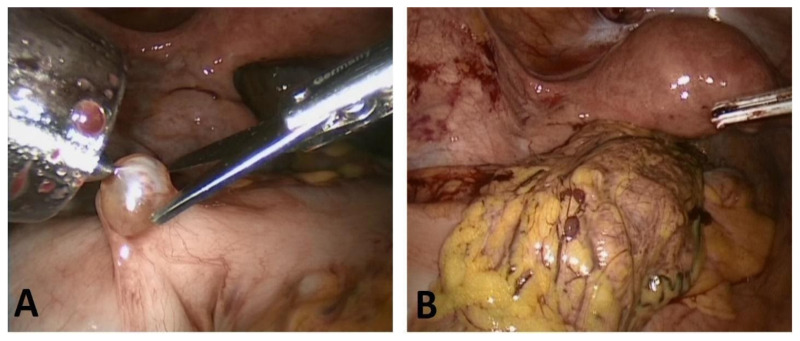
Macroscopic findings during laparoscopic surgery. Solid nodules were identified on the surface of the sigmoid colon (**A**) and omentum (**B**) (previously unpublished, original photos).

**Figure 6 diagnostics-12-01411-f006:**
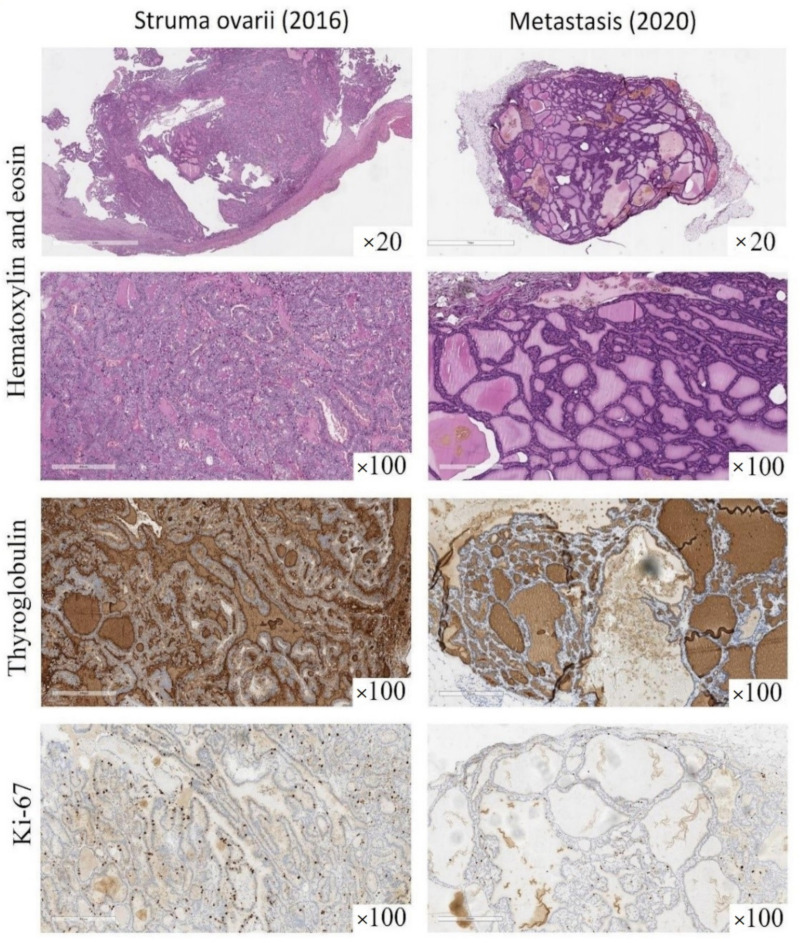
Morphology and immunohistochemical profile of ovarian struma and peritoneal deposits (previously unpublished, original photos). The primary ovarian neoplasm (**left column**) as well as the subsequent metastasis (**right column**) demonstrated macro- and micro-follicles containing colloid. The follicles were more densely packed in the primary ovarian neoplasm (“proliferative” struma), lacking the typical features of papillary or follicular carcinoma. In both specimens, tumor cells were diffusely positive for thyroglobulin immunostain (clone EPR 9730, rabbit monoclonal, dilution 1:500, Abcam, Cambridge, UK), while the Ki-67 labeling index was below 5% (clone 30-9, rabbit monoclonal, RTU; Ventana Medical Systems, Oro Valley, AZ, USA).

**Table 1 diagnostics-12-01411-t001:** Clinical cases of ovarian strumas: clinical course and morphological features.

Authors	Cases	Age	Benign	Atypical	Malignant	Metastasis	Treatment	Follow-Up after Last Treatment
Wei S. et al. [23]	96	median 46	80	0	16	liver, cul-de-sac, fallopian tube, urinary bladder, pelvic wall	-TR -LT4	1 month–20 years (NED/AWD)
Shaco-Levy R. et al. [2]	86	median 41	60	0	26	ovarian serosa, extraovarian spread	-TR -RAI	5–20 years (4 DOD, 82 NR)
Shaco-Levy R. et al. [24]	27	median 41	19	0	8	ovarian serosa, extraovarian spread (*n* = 17), including pelvis, omentum, peritoneum, liver, lung, bones	-SO ± HY ± tumor debulking -RAI (*n* = 12) -chemotherapy (*n* = 5)	1.5–33 years (3 NED, 9 AWD, 10 DOD, 5 DOC)
Devaney K. et al. [14]	54	mean 44	0	41	13	peritoneum (*n* = 1; at diagnosis)	-SO ± HY	2–18 years (NED)
Wang Y. et al. [25]	68	mean 42	64	0	4	3 malignant cases recurred	-TR -SO ± HY -THY/RAI (*n* = 2)	6 months–21 years (NED)
Marti J.M. et al. [26]	57	median 44	0	0	57	no recurrence	-SO ± HY -THY/RAI	6 months–25 years (NED)
Garg K. et al. [13]	10	median 41.5	0	0	10	uterine serosa, pelvic sidewall, cul-de-sac, diaphragm, omentum, liver	-TR -SO ± HY -THY/RAI	1–14 years (2 AWD, 8 NED)
Muallem M.Z. et al. [27]	1	38	1	-	-	paracolic gutter, left diaphragm, liver, spleen, gallbladder, omentum, ileocecal region, mesentery of the appendix, parametrium, right adnexa, pelvic peritoneum	-Right ovarian cystectomy -tumor debulking -THY/RAI	36 months (NED)
Ranade R. et al. [28]	1	55	1	-	-	liver, peritoneum, pelvic region, lugs, spleen, and bilateral adnexa (recurrence)	-SO + HY -THY/RAI	3 months (AWD)
Hwu D.-W. et al. [29]	1	28	1	-	-	bilateral ovaries, liver, peritoneum, lung	-SO + HY -THY/RAI	4 months (AWD)
Akahira J. et al. [5]	1	64	1	-	-	pelvic cavity, uterus, rectum, mesentery	-SO -TR	17 months (AWD)
Karseladze A.I. et al. [15]	1	49	1	-	-	contralateral ovary, omentum	-SO + HY + omentectomy -3 cycles of polychemotherapy	36 months (NED)
Riggs M.J. et al. [30]	1	32	1	-	-	anterior and posterior peritoneal reflections of the uterus	-SO + TH + pelvic resection -THY/RAI	36 months (AWD)
Oh S.-J. et al. [31]	1	60	1	-	-	T12 level of the spine	-SO + HY -TR -THY/RAI + spondylectomy	NR (AWD)
Zekri J. et al. [32]	1	26	1	-	-	lungs and skull (recurrence 120 months after surgery)	-SO -THY/RAI	60 months after RAI (AWD)
Kobayashi K. et al. [33]	1	39	1	-	-	bone (osteolysis at the Th7 level, ilium) (recurrence)	-oophorectomy -external beam radiotherapy (dose of 37.5 Gy for the spinal lesion)	9 months (NED)

AWD—alive with disease, DOC—dead of other cause, DOD—dead of disease, HY—hysterectomy, LT4—levothyroxine, NED—no evidence of disease, NR—not reported, RAI—radioactive iodine therapy, SO—salpingo-oophorectomy, THY—thyroidectomy, TR—Tumor resection.

## Data Availability

Not applicable.

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
