# Peer review of "Malignant Clinical Course of “Proliferative” Ovarian Struma: Diagnostic Challenges and Treatment Pitfalls"

_diagnostics, 2022, doi:10.3390/diagnostics12061411_

Round 1

Reviewer 1 Report

The authors have improved the content of the article.

Still, I have some suggestions. 

The abstract should be reorganized and focused on the particularities of the case.

The legend of the figures should include some relevant details.

The Conclusions section needs to be redone. The conclusions must be clear, concrete, and based in particular on the case report conducted by the authors.

In addition, they should carefully review the English language so as not to lose the quality of the information (e.g patients were usually alive...).

Author Response

Dear Reviewer,

Thank you for considering the manuscript Malignant Clinical Course of “Proliferative” Ovarian Struma: Diagnostic Challenges and Treatment Pitfalls (Special Issue Tissue-Based Biomarkers of Solid Tumors in the Routine Clinical Setting 2.0). 

We appreciate the time and effort that you dedicated to providing feedback on our manuscript and are grateful for the insightful comments on and valuable improvements to it. We have incorporated suggestions and highlighted changes within the manuscript. 

1) English language and style are fine/minor spell check required. The authors have improved the content of the article. ……..In addition, they should carefully review the English language so as not to lose the quality of the information (e.g patients were usually alive...).

ANSWER: Thank you so much. We have improved the manuscript according to the suggestions of the 2 Reviewers and the comments of the Editor, improving the English language.

2) The abstract should be reorganized and focused on the particularities of the case.

ANSWER: According to the suggestions of the Reviewer, we improved the abstract focusing on the particularities of our case. 

3) The legend of the figures should include some relevant details.

ANSWER: We have improved the legend of the figures according to the Reviewers’s suggestions.

4) The Conclusions section needs to be redone. The conclusions must be clear, concrete, and based in particular on the case report conducted by the authors.

ANSWER: According to your suggestions, we have improved the Conclusion.

Reviewer 2 Report

Reviewer's report

Manuscript ID: Diagnostics-1704998
Title:  Malignant Clinical Course of  Proliferative Ovarian Struma: Diagnostic Challenges and Treatment Pitfalls

Date: 2022/5/10

 Reviewer's report:
This is an interesting manuscript as it was a single case study of a highly rare disease –Ovarian Struma. Struma ovarii is a monodermal ovarian teratoma composed predominantly of thyroid tissue, a rare neoplasm comprising 1% of all ovarian tumors and 3% cases of ovarian teratoma. Morphologic features may or may not correlate with biologic behavior. This study is a report of a  rare case of ovarian struma that subsequently progressed with the development of peritoneal  and  osseous metastases four years after tumor resection. Until now, there are no large series of study of this disease, only a few numbers of case series were found in publication.

The MS is well prepared, the major limitation is a single case report. Nevertheless, its a novel study  which provide us a new perspective toward this rare disease. there's a few issue need to be answer  prior publication.

  1. Was there an image study of this disease ? If so what was the characteristic image of ovarian struma in CT scan or MRI?

Author Response

Dear Reviewer,

Thank you for considering the manuscript Malignant Clinical Course of “Proliferative” Ovarian Struma: Diagnostic Challenges and Treatment Pitfalls (Special Issue Tissue-Based Biomarkers of Solid Tumors in the Routine Clinical Setting 2.0). 

We appreciate the time and effort that you dedicated to providing feedback on our manuscript and are grateful for the insightful comments on and valuable improvements to it. We have incorporated suggestions and highlighted changes within the manuscript. 

1) I don't feel qualified to judge about the English language and style. This is an interesting manuscript as it was a single case study of a highly rare disease –Ovarian Struma. Struma ovarii is a monodermal ovarian teratoma composed predominantly of thyroid tissue, a rare neoplasm comprising 1% of all ovarian tumors and 3% cases of ovarian teratoma. Morphologic features may or may not correlate with biologic behavior. This study is a report of a rare case of ovarian struma that subsequently progressed with the development of peritoneal and  osseous metastases four years after tumor resection. Until now, there are no large series of study of this disease, only a few numbers of case series were found in publication. The MS is well prepared, the major limitation is a single case report. Nevertheless, its a novel study  which provide us a new perspective toward this rare disease. there's a few issue need to be answer  prior publication.

ANSWER: Thank you so much. We have improved the manuscript according to the suggestions of the 2 Reviewers and the comments of the Editor, checking the English language.

2) Was there an image study of this disease? If so what was the characteristic image of ovarian struma in CT scan or MRI?

ANSWER: Thank you for your comment. Only ultrasound examination of the ovarian struma was performed in 2016; according to your suggestions, we have added photos and information about it in the text. Additional imaging was not indicated at that time for an ovarian struma. However, the report of the surgical procedure (in 2016) did not describe peritoneal lesions. We have added a sentence about it in the text.

This manuscript is a resubmission of an earlier submission. The following is a list of the peer review reports and author responses from that submission.

Round 1

Reviewer 1 Report

The authors reported a rare case of a “borderline” ovarian struma with a malignant clinical course. The case presentation is presented in a well-structured manner.

However, some aspects need to be revised, as listed below:

Most likely the current secondary lesions diagnosed in 2020 (in bone, large omentum, sigmoid serosa) with malignant follicular morphology represent extraovarian disseminations of an initial malignant ovarian struma with the appearance of well-differentiated follicular carcinoma. Given that the 2016 ovarian cystic lesion had "macro and microfollicles containing colloid, as well as solid areas of pseudotubular structures, focal areas of cells with nuclear irregular contours, loss of polarity, and overlapping nuclei" leads us even further in an initial malignant component associated with the ovarian struma, even if the proliferation index was only 5% (although in Figure 4, Ki67 appears to be greater than 5% in the most active areas).

Considering the peritoneal implants, do you have data about frozen sections and peritoneal cytology?

It is well known that recurrences are less common in tumors limited to the ovary at the time of diagnosis, but even these patients require long-term follow-up. The CA 125 value should have been determined postoperatively and in dynamics. Thyroid hormones should have been dosed on a regular basis as well. Regular ultrasound examinations of the patient could have shown tumor recurrence in the ovary prior to the manifestation of bone metastases. Do you have data on these investigations?

Please explain why you requested TTF to rule out clear cell carcinoma? "Clear cell carcinoma was excluded due to immunoexpression of TTF-1." As far as it is known, among other indications, TTF1 indicates the thyroid nature of the struma in this case, rather than the clear cell phenotype.

I noticed that you did not consider a papillary thyroid carcinoma differential, although an HBME1 or Galectin 3 would help a lot to exclude this entity. Please explain.

Molecular testing has potential value in malignant ovarian struma. You mentioned “No BRAF mutations in the ovarian struma or metastatic lesions were detected. Consequently, we can assume that there is no molecular evolution between the primary ovarian struma and the metastatic lesions”. But BRAF testing is more useful for the papillary phenotype than for the follicular one. You have also considered additional tests such as RAS, PPARG, NRAS or other somatic mutations?

Why was only the left ovarian cystectomy and not the salpingo-oophorectomy performed following cesarean section in 2016? Salpingo-oophorectomy, thyroidectomy, and hysterectomy, to name a few typical therapies, might have been more appropriate in this case.

The article does not mention data on the reproductive status of the patient at the time of the second intervention in 2020, because this fact is important in deciding on fertility-conserving surgery for a patient with metastases. Do you have such information?

The article is well documented with references regarding the therapeutic decision and follow-up of other studies, but with a large number of cases. The evolution of these cases depending on the therapeutic decision should also been noted.

Conclusions should be more focused on the reported case.

Did you obtain the patient's consent for treatment or publication?

Thank you!

Reviewer 2 Report

The article has the typical shape of the case study.  The standard of the use of English is acceptable, and tables and all the figures are present.

The discussed case study is original and fills an important gap in the literature.

In the discussion section, literature is nicely cited, although: recent studies revealed that human chorionic gonadotropin can stimulate the TSH receptors of struma ovarii and potentially promote tumor growth during pregnancy. This aspect of pregnancy hormone should be expanded in the discussion part. I strongly recommend the article to explore the role of hCG : hCG – related molecules and their measurement (10.17772/gp/60981)